# Comparison of CRISPR–Cas9 Tools for Transcriptional Repression and Gene Disruption in the BEVS

**DOI:** 10.3390/v13101925

**Published:** 2021-09-24

**Authors:** Mark R. Bruder, Sadru-Dean Walji, Marc G. Aucoin

**Affiliations:** Department of Chemical Engineering, University of Waterloo, Waterloo, ON N2L 3G1, Canada; m2bruder@uwaterloo.ca (M.R.B.); dru.walji@gmail.com (S.-D.W.)

**Keywords:** CRISPR–Cas9, transcriptional repression, baculovirus, BEVS

## Abstract

The generation of knock-out viruses using recombineering of bacmids has greatly accelerated scrutiny of baculovirus genes for a variety of applications. However, the CRISPR–Cas9 system is a powerful tool that simplifies sequence-specific genome editing and effective transcriptional regulation of genes compared to traditional recombineering and RNAi approaches. Here, the effectiveness of the CRISPR–Cas9 system for gene disruption and transcriptional repression in the BEVS was compared. Cell lines constitutively expressing the *cas9* or *dcas9* gene were developed, and recombinant baculoviruses delivering the sgRNA were evaluated for disruption or repression of a reporter green fluorescent protein gene. Finally, endogenous AcMNPV genes were targeted for disruption or downregulation to affect gene expression and baculovirus replication. This study provides a proof-of-concept that CRISPR–Cas9 technology may be an effective tool for efficient scrutiny of baculovirus genes through targeted gene disruption and transcriptional repression.

## 1. Introduction

Baculoviruses are a diverse group of enveloped viruses that contain large, double-stranded DNA (dsDNA) genomes. The insect-specific infection cycle is complex and biphasic; primary infection of the midgut cells of the insect requires occlusion-derived virus (ODVs), which are virions encapsulated in a proteinaceous matrix, while cell–cell transmission is established by the budded virus (BV) phenotype [1]. Initial interest in baculoviruses nearly 100 years ago was for their promise as safe and selective bioinsecticides [2]; however, a recombinant baculovirus *Autographa californica* multiple nucleopolyhedrovirus (AcMNPV; rBEV) was used to produce human IFN-β in 1983, marking the inception of the baculovirus expression vector system (BEVS) as a platform for recombinant protein production [3]. Recombinant AcMNPV and its permissive cell lines Sf9 or High Five is the most common format of the BEVS [4]. A major milestone of BEVS biotechnology was the development of the bacmid system, which allowed site-specific integration of foreign genes in the AcMNPV genome propagated in *E. coli* [5]. Coupled with the λ-red or RecET recombineering systems [6,7], mutant AcMNPV genomes with sequence-specific deletions (i.e., knock-out viruses, KOVs) were efficiently obtained, allowing functional studies of many of the ~150 annotated open reading frames (ORFs) of AcMNPV [8,9,10,11,12,13]. Studies identifying *per os* infectivity factors, genes with insecticidal properties, and gene disruptions that improved recombinant protein production, have also been conducted using bacmid technology [14,15,16,17].

Bacmid technology has undeniably propelled baculovirus biotechnology, but like all genetic engineering approaches, it has drawbacks; even with recombineering, targeted gene disruption requires multiple experimental steps to achieve, and genetic instability has been reported during the ‘hyper-recombination’ state (i.e., during phage recombinase gene expression), resulting in intramolecular rearrangements in *E. coli* [18]. While this is largely abrogated by recombination of the bacmid and transfer vector in insect cells, isolation of the desired mutant rBEV is challenging with this approach [19]. The evaluation of KOVs is typically conducted by transfecting Sf9 cells with the bacmid and observing the spread of infection, which can be impacted by transfection efficiency, and it may fail to identify ORFs that are amenable to downregulation but not disruption. For example, the disruption of the AcMNPV ORF34 ORF resulted in its categorization as an essential gene; however, its downregulation by RNA interference (RNAi) not only permitted virus replication but improved heterologous gene expression [20]. Finally, trans-complementing cell lines may be required to enable the production of infective virions when essential genes are targeted for disruption [21,22].

Gene silencing via RNAi involves the production of double strand RNA (dsRNA) molecules that inhibit gene expression by triggering the degradation of messenger RNA (mRNA) through the RNA-induced silencing complex (RISC). The dsRNA molecules are typically introduced by the transfection of small interfering RNAs (siRNAs) directly or plasmids from which short hairpin RNAs (shRNA) are transcribed. While less prominently employed compared to gene disruption, RNAi has found utility in the BEVS [23]. Target sequence selection, however, often requires extensive empirical validation to identify high silencing efficiency targets, and the effectiveness may be limited by low transfection efficiency and relatively high cytotoxicity of the transfection reagent [24,25].

The CRISPR–Cas9 system has seen development for precision genome editing and targeted transcriptome engineering in a multitude of biological organisms over the past decade [26,27,28,29]. Genome editing was recently reported in Sf9 and High Five cells, and for genome editing of AcMNPV itself [30,31]. Here, we have extended upon these recent advancements by implementing and comparing strategies based on Cas9 and its nuclease deficient variant dCas9 for targeted gene disruption (CRISPRd) and transcriptional repression (CRISPRi), respectively. Using cell lines developed for expression of the *cas9* or *dcas9* genes and sgRNAs delivered by the rBEV, 4 genes on the rBEV were targeted for gene disruption and transcriptional repression. In each case, the expected phenotype was observed. Importantly, virus seed stocks could be produced in parental Sf9 cells, and the same rBEVs could be used for evaluating gene disruption and transcriptional repression. Target gene selection is achieved by exchanging the spacer sequence of the sgRNA, which can be done using conventional PCR. Accordingly, this approach simplifies the generation of KOVs by reducing the experimental steps required and allows for investigation of gene function using gene disruption and transcriptional repression using the same rBEV.

## 2. Materials and Methods

### 2.1. Cells

Sf9 cells were maintained in suspension culture in Gibco SF900 III serum-free medium (Fisher Scientific, Whitby, ON, Canada) as described previously [32]. Sf9 cells were transfected as adherent cultures in tissue culture treated 6-well plates (VWR, Mississauga, ON, Canada) with Escort IV transfection reagent (Sigma-Aldrich, Oakville, ON, Canada) according to manufacturer’s directions. To derive the transgenic Sf9-Cas9 and Sf9-dCas9 cell lines, parental Sf9 cells were transfected with the plasmid pOpIE2-Cas9-puro or pOpIE2-dCas9-puro, respectively. Approximately 48 h post transfection (hpt), growth medium was aspirated and replaced with fresh medium containing 5 µg/mL puromycin (Sigma-Aldrich). Selective pressure was maintained for at least 2 weeks, and resistant cells were pooled and adapted back to suspension culture. The recombinant cells were maintained in suspension culture for ~10–15 additional passages in medium containing puromycin prior to further analysis.

### 2.2. Plasmid Construction

All plasmids used in this study were constructed using NEBuilder HiFi DNA Assembly master mix (New England Biolabs, Whitby, ON, Canada) according to the manufacturer’s directions. Primers used for construction of all plasmids and retargeting sgRNAs were purchased from Integrated DNA Technologies (IDT; Coralville, IA, USA) and are listed in Appendix A.

To construct plasmid pOpIE2-Cas9-puro, the cas9-T2A-pac region from pAc-sgRNA-Cas9 [33] (Addgene #49330, Cambridge, MA, USA) and a fragment containing the *Orgyia pseudotsugata* MNPV immediate-early 2 promoter (OpIE2) and 3’ untranslated region (UTR) [34], origin of replication (ori), and ampicillin resistance gene (ampR) for propagation in *E. coli* were amplified via PCR, and the 2 PCR fragments were used in a Gibson assembly reaction [35]. The resulting plasmid placed the cas9-T2A-puro expression cassette under the control of the constitutive OpIE2 promoter. To generate plasmid pOpIE2-dCas9-puro, the dcas9 ORF was amplified from pdCas9::BFP-humanized [28] (Addgene #44247) and used in a Gibson assembly reaction along with a PCR fragment containing the OpIE2 promoter, T2A-puro cassette, and OpIE2 3’ UTR to place the dCas9 gene under the constitutive control of the OpIE2 promoter.

For plasmids containing the OpIE2-GFP cassette and SfU6 sgRNA, the *mAzami-Green* gene (Addgene #54798; herein referred to as *gfp*) encoding a monomeric green-emitting fluorescent protein [36] gene was PCR amplified and placed between the OpIE2 promoter region and 3’ UTR. Separately, the *S. frugiperda* U6-3 (SfU6) small nuclear RNA (snRNA) promoter [30] was synthesized as a gblock (IDT), and PCR amplified along with the single guide RNA (sgRNA) and transcriptional terminator from plasmid pCFD4-U6:1_U6:3tandemgRNAs [37] (Addgene #49411). The OpIE2-GFP fragments were inserted along with the SfU6-sgRNA DNA fragment into pACUW51 to derive plasmid pOpIE2GFP-sgRNA.

To construct p10-GFP and p6.9-GFP-encoding CRISPR plasmids, first, the coding region of the *p10* gene, including upstream and downstream sequences to include its endogenous promoter and 3’ UTR, was amplified from AcMNPV genomic DNA and inserted into pACUW51. The *p10* ORF was then replaced with the *gfp* gene, and the SfU6-sgRNA fragment was inserted downstream to derive p10GFP-sgRNA. Finally, the p6.9 promoter region was amplified from AcMNPV genomic DNA and inserted in place of the p10 promoter sequence in p10GFP-sgRNA to yield p6.9GFP-sgRNA.

The spacer sequences used to target Cas9 and dCas9 to specific AcMNPV genomic loci were selected using the sgRNA scorer 2.0 software [38]. Briefly, the coding sequence for the target gene was submitted to the sgRNA scorer 2.0 software which generated a list of putative target sites scored according to their predicted activity. For each gene target, 2–4 target sequences were selected based on two criteria: predicted activity and the strand (template or nontemplate) the target sequence resided on. Inverse PCR was used to retarget sgRNA spacer sequences to the target of interest [39]. Two primers were designed to anneal to the cas9 handle of the sgRNA sequence and to the U6 promoter sequence and extend in opposite directions. The desired targeting spacer sequence was appended to these primer sequences, which were used to amplify the entire plasmid as a linear fragment. The spacer sequence served as the homologous sequence required for Gibson assembly to ligate and re-circularize the new plasmid. The spacer sequences used in this study are presented in Table 1.

### 2.3. Virus Generation, Amplification, and Quantification

Plasmids for homologous recombination at the polyhedrin locus in the AcMNPV genome were co-transfected with flashBACGOLD™ (Oxford Expression Technologies Ltd., Oxford, UK) genomic DNA according to manufacturer’s directions. Supernatant from each transfection was harvested 4–5 days post transfection and used to infect early exponential phase (~1.5–2 × 10^6^ cells/mL) suspension Sf9 cultures at low multiplicity of infection (MOI; MOI = ~0.01–0.1) to amplify the rBEVs for 3–4 days or until the viable cell density dropped to ~80%. After 2 sequential rounds of amplification, the rBEV titer was quantified using end-point dilution assay (EPDA). Briefly, Sf9 cells were diluted to ~2.0 × 10^5^ cells/mL, and 100 µL was used to seed each well of a 96-well plate (VWR). The virus was serially diluted (10^−2^ to 10^−8^), and 10 µL of each dilution was added, in 12 replicates, to the 96-well plate. Plates were incubated for seven days at 27 °C, after which they were checked for green fluorescence using a fluorescence microscope. Results were converted from TCID_50_ and reported as plaque forming units per ml (pfu/mL). Where appropriate, tests for statistical significance of differences in infectious virus titer were performed using Student’s *t*-test, with a significance level value of 5%.

### 2.4. Infections

Sf9-dCas9, Sf9-Cas9, or the parental Sf9 cells were infected with rBEVs at a density of ~1.5–2 × 10^6^ cells/mL and MOI of 3. Samples were taken at 24, 48, and 72 h post infection (hpi), and cells were fixed with 2% paraformaldehyde for ~30 min prior to further analysis.

### 2.5. Flow Cytometry and Analysis

Samples were analyzed using a FACSCalibur™ flow cytometer (BD Biosciences, San Jose, CA, USA) equipped with a 15 mW air-cooled argon-ion laser with an excitation frequency of 488 nm. Samples were run at the low flow setting (12 µL/min), and 10,000 events were collected. The analysis of flow cytometry data was performed using FlowJo^®^ V10 flow cytometry analysis software (FlowJo LLC, Ashland, OR, USA). Briefly, after applying gates to remove debris and intrinsic cellular fluorescence from the analysis, median fluorescence intensity in FL1 was calculated. Where appropriate, tests for statistical significance of differences in fluorescence intensity were performed using Student’s *t*-test, with a significance level value of 5%.

### 2.6. Real-Time Reverse Transcription Polymerase Chain Reaction (RT-PCR)

Infected cells (~1.5–2 × 10^6^ cells/mL, MOI = 3) were collected at 0, 24, 48, and 72 hpi by centrifugation at 1000× *g* for 10 min at 4 °C. RNA was extracted using the Geneaid Total RNA Mini kit (FroggaBio, Concord, ON, Canada), and 500 ng was used as template for first-strand cDNA synthesis using the SensiFAST cDNA synthesis kit (FroggaBio) according to manufacturer’s directions. Real-time PCR was performed using the SensiFAST SYBR Hi-ROX kit (FroggaBio) according to manufacturer’s directions on an Applied Biosystems StepOnePlus™ Real-Time PCR System (Fisher Scientific). Primer pairs used are given in Table S1.

### 2.7. Western Blot

Infected cells (~1.5–2 × 10^6^ cells/mL, MOI = 3) were collected at 0, 24, 48, and 72 hpi by centrifugation at 1000× *g* for 10 min at 4 °C. The cells were lysed in RIPA buffer (Fisher Scientific), quantified by BCA assay (Fisher Scientific), and ~10 µg of protein was separated by electrophoresis in 10% TGX Stain-Free precast mini SDS-PAGE gels (Bio-Rad, Mississauga, ON, Canada) according to manufacturer’s directions. After transfer to PVDF membranes, Western blot analysis was performed with anti-Cas9 (MAC133; Sigma-Aldrich) or anti-GP64 (AcV5, Fisher Scientific) as primary antibodies and goat anti-mouse IgG HRP secondary (Bio-Rad) and imaged on a ChemiDoc MP Imager (Bio-Rad). The Image Lab software was used for further image processing (Bio-Rad).

### 2.8. Quantification of Baculovirus Particles Using Flow Cytometry

Sample preparation for analysis via flow cytometry was described previously [40]. Briefly, samples were diluted in D-PBS and fixed with paraformaldehyde for ~1 h, after which the samples were subjected to one freeze–thaw cycle followed by incubation with Triton X-100 to permeabilize the membrane. The nucleic acid stain SYBR Green I was added and incubated at 80 °C for 10 min in the dark to stain double stranded DNA. After cooling on ice, the samples were analyzed via flow cytometry. Flow-Set Fluorospheres (Beckman Coulter, Mississauga, ON, Canada) were used for calibration, and all samples were run in triplicate. The statistical significance was evaluated using Student’s *t*-test with a significance value of 5%.

## 3. Results

### 3.1. Development of Recombinant Sf9 Cell Lines for Constitutive Expression of Cas9 and dCas9

Expression of the Cas9 and dCas9 proteins were conferred via the development of transgenic Sf9 cell lines transfected with plasmids pOpIE2-Cas9-puro and pOpIE2-dCas9-puro, which include either the *cas9* or *dcas9* gene and the *pac* gene sequences separated by the viral T2A element [41]. After selection with puromycin for at least 2 weeks, resistant cells were pooled and maintained in suspension culture for an additional ~10–15 passages under selective pressure before removing the puromycin from the medium. Although routine maintenance of these cell lines with or without puromycin provided no evidence that ectopic expression of either Cas9 protein had any effect on their growth (data not shown), prior to performing any gene disruption or transcriptional repression experiments, the cell lines were characterized with infection experiments to determine whether there were any distinguishable differences between them and parental Sf9 cells. As shown in Figure 1A, transcription of the *gfp* reporter and the viral capsid protein *vp39* genes were similar, indicating that there were no discernable differences in progression of the infection. Similarly, the production of GFP from the viral late gene promoter p6.9 and progeny virus appeared unimpaired (Figure 1B,C). Interestingly, RT-PCR (Figure 1A) and Western blot data (Appendix A) indicated that transcription of *cas9* and *dcas9* were significantly downregulated by 24 hpi and protein was undetectable on Western blot by 48 hpi.

### 3.2. Evaluation of CRISPR-Mediated Repression and Disruption on GFP Production

Initial experiments sought to establish transcriptional repression of the rBEV-encoded *gfp* gene. Individual rBEVs with sgRNAs targeting the template (GFP1 and GFP4) and nontemplate (GFP2 and GFP3) strands within the *gfp* ORF were constructed, and repression of *gfp* transcribed with immediate early (OpIE2), late (p6.9), and very late (p10) promoters was assessed. Sf9-dCas9 cells infected with rBEVs encoding nontemplate strand-targeting sgRNAs showed a marked decrease in the proportion of GFP-positive cells and fluorescence intensity compared to the control at 48 hpi for OpIE2-GFP (Appendix A). For p6.9-GFP and p10-GFP, there appeared to be a slight (p6.9-GFP) or significant (p10-GFP) reduction in GFP-positive cells compared to the control at 24 hpi; however, there was no difference at 48 and 72 hpi (Figure 2A and Appendix A). Nevertheless, the fluorescence intensity for GFP2 and GFP3 targets was reduced compared to the control at all time points for p6.9-GFP and at 48 and 72 hpi for p10-GFP. The fluorescence intensity of rBEVs encoding strand targeting sgRNAs (GFP1 and GFP4) were indistinguishable from the control in all experiments, however, indicating potential strand bias for CRISPRi.

For CRISPRd experiments, p6.9-GFP rBEVs encoding sgRNAs GFP2, GFP3, and GFP4 were used to infect Sf9-Cas9 cells. For all three sgRNAs, the proportion of GFP-positive cells was significantly reduced compared to the control at all time points. The GFP2 sgRNA resulted in the lowest GFP-positive phenotype compared to GFP3 and GFP4. Significantly, whereas the proportion of GFP-positive cells was higher at 24 hpi and increased by 48 hpi for GFP3 and GFP4, GFP2 was less than 10% GFP-positive at 24 hpi and did not increase as the infection progressed. For the fluorescence intensity measurements of the GFP-positive cells, though, the GFP2 sgRNA rBEV was only slightly reduced compared to the untargeted control rBEV. Conversely, the GFP3 and GFP4 sgRNAs had significantly reduced fluorescence intensity compared to both GFP2 and control (Figure 2C,D). 

Importantly, parental Sf9 cells infected with each of the p10-GFP (data not shown) and p6.9-GFP rBEVs produced fluorescence intensity measurements that were both increased compared to the same infections in either Sf9-Cas9 or Sf9-dCas9 cells and were also all similar to the control (i.e., untargeted) rBEV. Finally, the release of progeny BV was not statistically different for any of the viruses replicating in any cell line (Appendix A). Taken together, these data indicate that the production of GFP was influenced by the presence of both (d)Cas9 and sgRNA and was unaffected in the absence of either of these molecules, and is therefore consistent with CRISPR-mediated targeting of the *gfp* gene.

### 3.3. Extension of CRISPRi and CRISPRd to Endogenous AcMNPV ie-1 and vlf-1

Next, the ability of the CRISPRi and CRISPRd systems to affect the production of endogenous AcMNPV genes was assessed (Figure 3). Spacer sequences were selected to target the *ie-1* and *vlf-1* genes encoding immediate-early protein 1 (IE-1) and very late factor 1 (VLF-1), respectively. The *ie-1* gene encodes a transcriptional activator and is essential for viral DNA replication, late gene expression, and subsequent progeny virus production. The vlf-1 gene encodes a transcriptional activator for the very late class of genes but has no effect on late gene promoters. The production of progeny virus and GFP transcribed from the p10 promoter was measured to assess the phenotypic impact of these targets in Sf9-dCas9 cells. Similar to previous experiments, the proportion of GFP-positive cells was reduced at 24 hpi for the IE1 sgRNA; however, it increased by 48 hpi and was indistinguishable from the control. The proportion of GFP-positive cells was not affected for either IE2, VL1, or VL2. Reduced fluorescence for the rBEVs encoding nontemplate-targeting sgRNAs (IE1 and VL1), but not the sgRNAs targeting the template strand (IE2 and VL2) was observed. Finally, the analysis of progeny virus production showed that the IE1 sgRNA reduced the infectious virus titer (IVT) ~90% compared to the control at 48 hpi. The difference in IVT for the other targets was not statistically significant (Figure 3A).

For CRISPRd in Sf9-Cas9 cells, a marked reduction in GFP-positive cells was observed for both IE1 and IE2 sgRNA rBEVs, but not VL1 or VL2, at all time points. Fluorescence intensity was also significantly reduced for each sgRNA compared to the untargeted control. Since VLF-1 stimulates transcription from very late promoters but has no effect on late genes, rBEVs encoding the VL1 and VL2 sgRNAs and the p6.9-GFP expression cassette were prepared and used to infect Sf9-Cas9 cells. Importantly, the analysis of fluorescence indicated no difference compared to the control rBEV for both VL1 and VL2 (data not shown). Finally, the IVT was reduced by ~99% compared to the control for IE1 and IE2, and ~64% for VL2. The measured IVT for VL1 was not statistically different from that of the control (Figure 3B).

Parental Sf9 cells infected with each of *ie-1* and *vlf-1*-targeted sgRNAs showed fluorescence and IVT levels that were consistent with those of the control (i.e., nondisrupted/repressed), indicating that both (d)Cas9 and sgRNA are required for disruption or downregulation of *ie-1* and *vlf-1* (Figure 3C). Additionally, since both fluorescence intensity and IVT were similar for all the rBEV infections in Sf9 cells, it is unlikely that the phenotypes observed in Sf9-Cas9 or Sf9-dCas9 are the result of genetic heterogeneity between the rBEVs or differences in late gene expression and/or virus replication.

### 3.4. CRISPRd Is More Effective Than CRISPRi for Obstructing Progeny BV Release

Finally, rBEVs with sgRNAs targeting the vp80 gene were prepared. The vp80 gene encodes the capsid-associated protein VP80, and its disruption prevents capsid assembly but has no effect on late gene expression. The fluorescence intensity of GFP transcribed from the late p6.9 promoter and progeny virus production were used to assess the effectiveness of CRISPRi and CRISPRd, and four sgRNAs were tested for CRISPRi and two for CRISPRd. The fluorescence intensity was similar for each vp80-targeting rBEV compared to the untargeted control in each cell line, and showed no differences when used to infect either Sf9, Sf9-Cas9, or Sf9-dCas9 cells (Figure 4A). Similarly, the IVT of infected Sf9 cell culture supernatants for each rBEV were similar, indicating unimpaired BV release in the absence of Cas9 or dCas9. The VP2, VP3, and VP4 rBEVs reduced IVT ~79%, ~68%, and ~57% compared to the control rBEV, respectively, in Sf9-dCas9 cells, while VP1 was similar to the control and to the IVT yielded from its infection of parental Sf9 cells. Infection with VP1 to Sf9-Cas9 cells, on the other hand, reduced the IVT by ~98% compared to the untargeted control, and VP2 by ~96% (Figure 4C). The latter result represents an ~85% improvement over the result in Sf9-dCas9 cells. Finally, flow cytometry analysis indicated a reduction in total number of particles in culture supernatant for VP80 targets as compared to nontargeted control experiments in Sf9-Cas9 cells (Figure 4B).

## 4. Discussion

The present study sought to develop an efficient technology for targeted genome engineering that would be capable of scrutinizing the effect of gene disruption or repression on viral gene expression and replication. To this end, transgenic Sf9 cell lines constitutively expressing the *cas9* or *dcas9* gene were developed. To test the efficacy of gene disruption and transcriptional repression using this approach, Sf9-Cas9 and Sf9-dCas9 cells were infected with rBEVs encoding both the sgRNA and the genetic target for disruption or repression. 

Since downregulation of host protein expression due to infection with AcMNPV is a characteristic of the BEVS [42], experiments to assess expression of (d)Cas9 using RT-PCR and Western blot were conducted. Consistent with prior studies, downregulation of *dcas9* and *cas9* transcription was observed by 24 hpi; however, the baculovirus immediate-early promoter supported transgene expression until 72 hpi [43,44]. As the amount of dCas9 protein present could impact the effectiveness of CRISPRi, the repression of *gfp* transcribed from immediate-early (OpIE2), late (p6.9), and very late (p10) promoters were evaluated to establish the efficiency of repression for promoters differing in temporal and relative strength expression characteristics. A significant reduction in fluorescence intensity was observed for each promoter. Additionally, the possibility of a ‘strand bias’ was observed in the BEVS system, in which transcriptional repression can only be achieved by targeting the sgRNA/dCas9 RNP to the nontemplate strand. This phenomenon has been observed in various other prokaryotic and eukaryotic systems previously [26,28]. The experiments conducted with Sf9-Cas9 cells (CRISPRd) showed decreased fluorescence intensity measurements compared to the control, and the population of GFP-positive cells was significantly reduced. Notably, GFP2 was less than 5% GFP-positive; however, fluorescence intensity was higher in Sf9-Cas9 cells than in Sf9-dCas9 cells with the same rBEV. This observation is presumably due to the mechanisms by which CRISPRd and CRISPRi function; for CRISPRi, successful targeting blocks transcript elongation and leads to a reduction in mRNA produced and translated by the cell. This ultimately leads to an overall, but typically incomplete, reduction in fluorescence intensity without introducing irreversible mutations in the genome. On the other hand, gene disruption mediated by CRISPRd results in indel mutations from dsDNA break repair, which are irreversible and often result in loss-of-function mutations in the encoded protein [28,29]. Consequently, gene copies with indel mutations in the ORF are transcribed, but the encoded protein is nonfunctional, whereas gene copies that are not successfully targeted or the indel mutation is silent would be transcribed and function protein is translated at wildtype levels.

Finally, endogenous AcMNPV *ie-1*, *vlf-1* and *vp80* genes were targeted for transcriptional repression and gene disruption. The IE-1 protein is the major transcriptional regulator of AcMNPV and is responsible for *trans*-activation of several known early genes [45,46]. Importantly, it is required for late gene expression and viral genome replication [47], and its deletion results in loss of infectivity [11]. The VLF-1 protein is a regulator of very late gene transcription and is responsible for the ‘transcriptional burst’ observed for the very late class of genes but has no effect on the late class of genes [48,49]. Complete deletion of the *vlf-1* gene may also impair assembly of BVs, although DNA replication and late gene transcription appeared to be reduced but permitted [13]. On the other hand, the *vp80* gene encodes a capsid-associated protein that is essential for BV production but not for viral late gene expression [21]. Selection of these endogenous genes provided the ability to observe the efficacy of CRISPRi and CRISPRd in several ways; repression/disruption of *ie-1* should impact the entire infection cycle of the rBEV, while targeting *vlf-1* should reduce expression from the very late p10 promoter but not the p6.9 promoter. Finally, disruption of *vp80* expression should impact the production of progeny BV but not inhibit late gene expression. 

Infections with rBEVs encoding sgRNAs targeting each of these genes yielded the expected result in all three cases; significant reduction in GFP and BV release for IE-1 targets, reduced BV production but unimpaired late gene expression for VP80 targets, and targeting VLF-1 led to a reduction in fluorescence intensity for GFP expressed from the p10 promoter but not p6.9. Interestingly, although there appears to be a reduced IVT for VL1 and VL2 rBEVs in Sf9-Cas9 cells (~42% and ~64%, respectively) and for VL1 in Sf9-dCas9 cells (~50%), only the VL2 rBEV IVT was statistically different from the control. This could indicate that either the resolution in the assay is not sensitive enough to detect this difference or that enough VLF-1 was produced to permit replication and production of progeny virus to near wildtype levels. Unsurprisingly, the template-targeting sgRNA VP1 did not result in reduced progeny virus production in Sf9-dCas9 cells. Targeting the *vp80* gene with sgRNAs VP2, VP3, and VP4, however, reduced IVT by ~79%, ~68%, and ~57%, respectively; however, the result with VP4 was not statistically significant. Given that transcriptional repression efficiency has been observed to be inversely correlated to the distance of the target spacer sequence from the transcriptional start site [28], it may not be entirely surprising that this sgRNA was less effective.

In addition to the apparent strand bias in the experiments with Sf9-dCas9 cells, the proportion of cells displaying a GFP-positive phenotype at 24 hpi was substantially reduced for the IE1 rBEV, while fluorescence was significantly lower at all time points and for both targets in Sf9-Cas9 cells compared to control infections. The analysis of BV release at 48 hpi also showed a ~90% decrease in the IVT for IE1 compared to controls for CRISPRi and ~99% for CRISPRd. This latter result is significant since a report in which transformed Sf9 cells expressing a ~470 bp dsRNA molecule targeting the AcMNPV *ie-1* gene exhibited strong viral repression at early stages of infection, but subsequent recovery of viral proliferation was observed by the late stages of the infection cycle [50].

Deletion of the *vp80* gene has previously been shown to prevent BV production whilst permitting replication of viral DNA and transcription of viral late genes at or near wildtype levels [21]. The results presented here support these conclusions: production of GFP was similar for each virus in Sf9-dCas9, Sf9-Cas9 and the parental Sf9 cells; however, BV release was decreased by >90% in Sf9-Cas9 cells. Interestingly, the supernatant from Sf9 cells infected with the Δ*vp80*-rBEV in that study appeared to have undetectable IVT [21]. The assessment of infected culture supernatants at 4, 8, and 12 hpi here and previously [32], however, revealed IVT ~10^4^–10^5^ pfu/mL at each time point (data not shown), indicating incomplete viral uptake before the onset of progeny BV release. Further, the *trans*-complementation strategy resulted in a ~25-fold decrease in BV seed production and constitutive expression of the *vp80* gene appeared unstable or toxic to Sf9 cells. Finally, higher MOI (MOI = 10) was required to produce recombinant protein at the same level as the wildtype rBEV [21]. In this study, there was no difference in GFP production at MOI = 3 and each of the rBEVs displayed no indication of impaired replication in Sf9 cells. Taken together, this strategy may contribute to reduced downstream processing complexity by minimizing rBEV contamination. Nevertheless, targeted disruption of *vp80* reduced the IVT by ~98% and ~96% for VP1 and VP2, respectively. Compared to CRISPRi, these results indicate that CRISPRd may be more effective for reducing progeny virus production. Finally, to ensure that targeting *vp80* resulted in a reduction in particles released to the culture supernatant as opposed to the release of defective particles that are not infectious, flow cytometry was used to analyze supernatants from several control and VP80-targeted infections in Sf9-Cas9 cells. The results indicated a ~90% reduction in particle concentration in the VP80-disrupted infections compared to the control. Consistent with previous reports in which the ratio of total particles quantified using flow cytometry to IVT measured using EPDA ranged from 1 to 10 [40], the FC:IVT ratio in the samples analyzed was ~5–10 as well.

## 5. Conclusions

Taken together, the phenotypes observed in this report are consistent with disruption or repression of the endogenous AcMNPV *ie-1*, *vlf-1*, *vp80* genes. The results indicate that CRISPRd may be more effective than CRISPRi for total disruption of target genes, whereas CRISPRi allows expression of the targeted gene at levels that are lower than the wildtype, suggesting it may be more appropriate for targets that are not amenable to deletion. Consequently, the CRISPRd tool developed here may be more useful for evaluating the essentiality of endogenous AcMNPV genes and reducing BV contamination in culture supernatants, whereas CRISPRi may be more effective for use in prolonging the infection cycle and accompanying bioprocess in order to increase yield of the target recombinant molecule. This report serves as a foundation for further improvement of the BEVS as a platform for recombinant protein therapeutics.

## Figures and Tables

**Figure 1 viruses-13-01925-f001:**
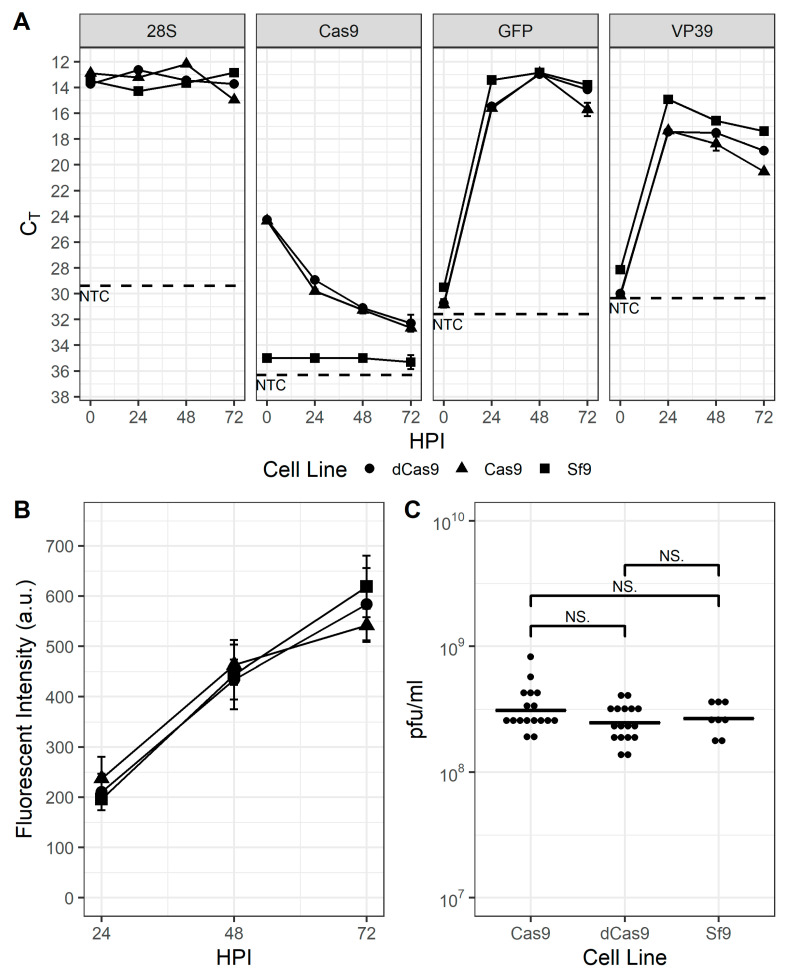
Sf9-Cas9 and Sf9-dCas9 cells are indistinguishable from the parental Sf9 cell line. (**A**). QPCR threshold cycle (C_t_)values of the Sf9-encoded 28S rRNA housekeeping gene, *(d)cas9* gene, and virus-encoded *vp39* and p6.9 promoter-expressed *gfp* reporter gene in baculovirus-infected Sf9 (squares), Sf9-Cas9 (circles), and Sf9-dCas9 (triangles) cells at 0, 24, 48, and 72 hpi. NTC: no template control (**B**). GFP fluorescence intensity of baculovirus-infected Sf9 (squares), Sf9-Cas9 (circles), and Sf9-dCas9 (triangles) cells at 24, 48, and 72 hpi and (**C**). infectious virus titer of supernatants from infected Sf9, Sf9-Cas9, and Sf9-dCas9 cells at 72 hpi. NS indicates non-significant differences in means according to Student’s *t*-test.

**Figure 2 viruses-13-01925-f002:**
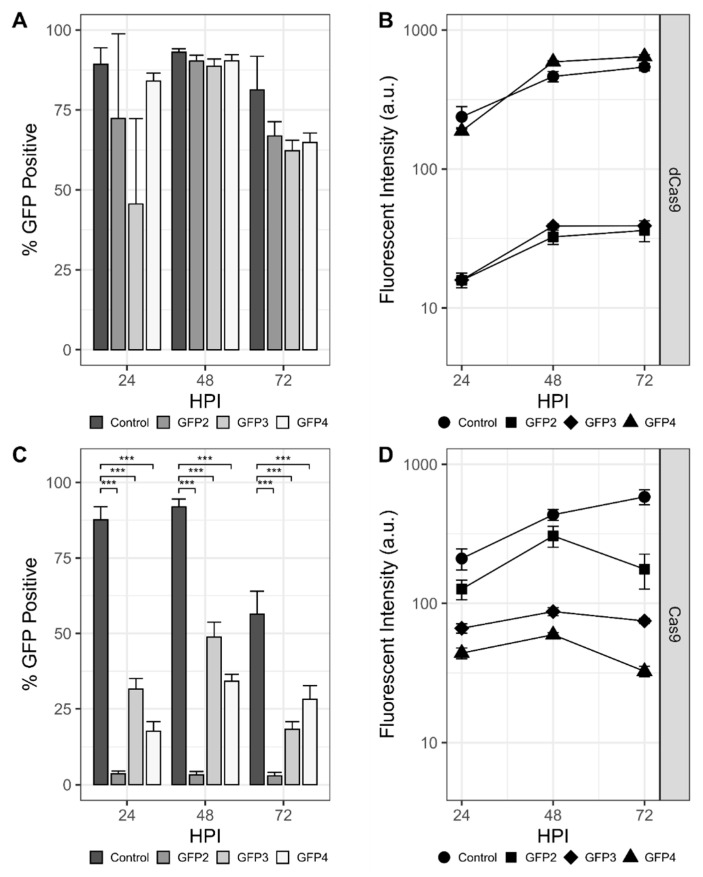
CRISPR-mediated targeting of *mAzami-Green* (*gfp*) transcribed from the late p6.9 promoter. Percentage of the population that is GFP-positive and fluorescence intensity of *gfp*- targeting rBEVS in Sf9-dCas9 cells (**A**,**B**) and Sf9-Cas9 cells (**C**,**D**), respectively. *** denotes *p* < 0.001.

**Figure 3 viruses-13-01925-f003:**
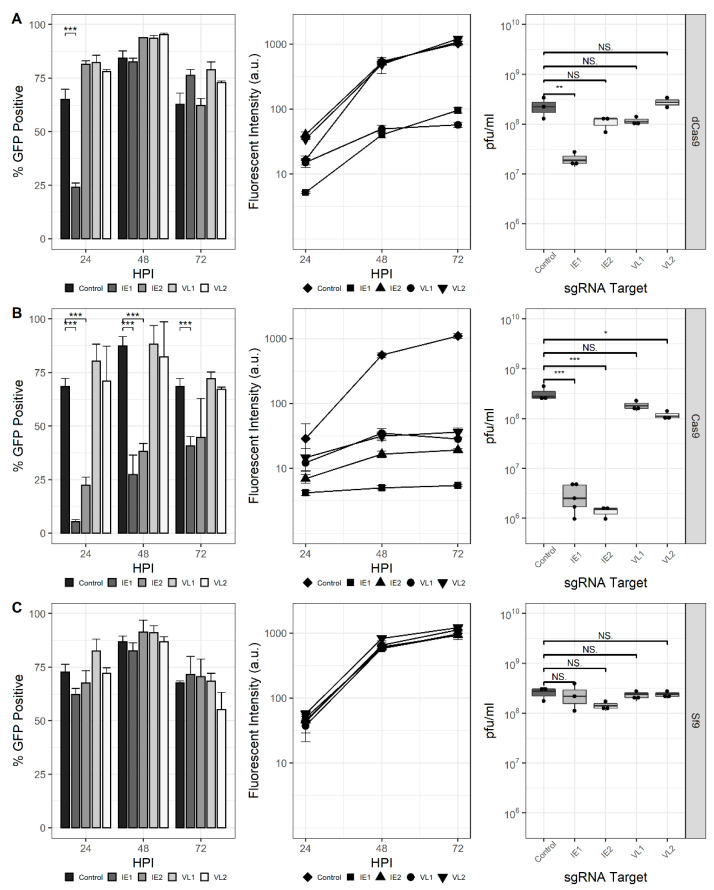
CRISPR-mediated targeting of the AcMNPV *ie-1* and *vlf-1* genes. Percent GFP-positive, fluorescence intensity, and IVT for rBEVs in (**A**) Sf-dCas9, (**B**) Sf9-Cas9, and (**C**) parental Sf9 cells. ***: *p* < 0.001; **: *p* < 0.01; *: *p* < 0.05; NS: non-significant.

**Figure 4 viruses-13-01925-f004:**
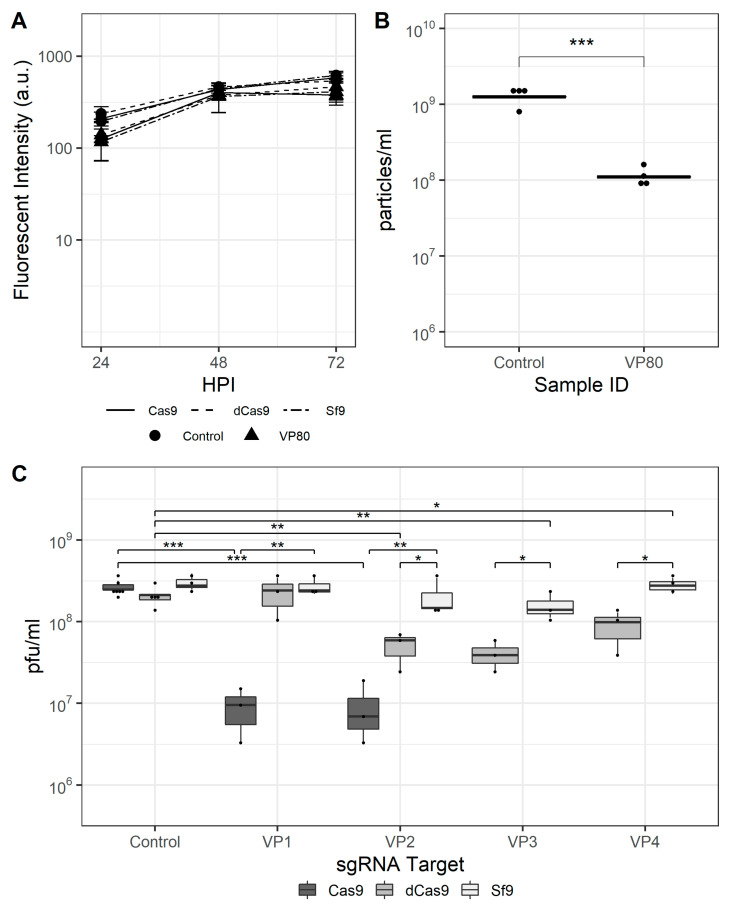
CRISPR-mediated targeting of the AcMNPV *vp80* gene. (**A**) Mean fluorescence intensity for *vp80*-targeting and control rBEVs in Sf9-Cas9, Sf9-dCas9, and parental Sf9 cells. (**B**) Total particles in culture supernatants of infected Sf9-Cas9 cells and (**C**) IVT for control and *vp80*-targeting rBEVs in each cell line at 48 hpi. ***: *p* < 0.001; **: *p* < 0.01; *: *p* < 0.05.

**Table 1 viruses-13-01925-t001:** Protospacer sequences for sgRNA targets.

rBEV	Target	Protospacer Sequence (5’–3’)	PAM	Strand
Control	n/a	caccttgaagcgcatgaact	n/a	n/a
GFP1	*gfp*	gggcaagggcaacccctacg	agg	Template
GFP2	*gfp*	gtcgtaggcgaagggcaggg	ggg	Nontemplate
GFP3	*gfp*	gttgccgtactggaacacgg	tgg	Nontemplate
GFP4	*gfp*	ccgagggctaccactgggag	agg	Template
IE1	*ie-1*	accgtgtcggctccatccgggg	tgg	Nontemplate
IE2	*ie-1*	tgatatctgacagcgagactg	cgg	Template
VL1	*vlf-1*	acacggactcgaaccggggag	cgg	Nontemplate
VL2	*vlf-1*	ggcaacgatgcacgcccgacg	agg	Template
VP1	*vp80*	gcccgccgcaatcgccgccg	cgg	Template
VP2	*vp80*	gctggatgttacccgcgg	cgg	Nontemplate
VP3	*vp80*	tcgatgcggccaggtcgc	tgg	Nontemplate
VP4	*vp80*	gcggatcgctaaatgccg	tgg	Nontemplate

## Data Availability

All data generated and analyzed during this study are included in this article and its Appendix A.

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
