# Peer review of "Comparison of CRISPR–Cas9 Tools for Transcriptional Repression and Gene Disruption in the BEVS"

_viruses, 2021, doi:10.3390/v13101925_

Round 1
Reviewer 1 Report
Comparison of CRISPR-Cas9 tools for transcriptional repression and gene disruption in the BEVS
In this work, Bruder et al showed that Cas9 may be more effective than dCas9 for total gene disruptions, whereas dCas9 allows gene expression albeit at reduced level relative to the wildtype. Bruder et al used the Baculovirus as a model organism for the study.
This is an interesting study with good experimental plans. Adequate controls were employed to clearly show the impacts of the CRISPR/Cas9 tools. The manuscript is well written in a simple-to-understanding language without compromise to scientific facts/data. Findings from this study will be useful to researchers using Baculovirus vector systems (BEVs)
Other comments
Line 81: “serum-free” not “serum free”
Lines 67-70: This study no doubt makes an original contribution to the field of BEVs. However, I find the statement “Here, we have extended upon these recent advancements by developing engineering tools based on Cas9 and its nuclease deficient variant dCas9 for targeted gene disruption…” quite an over statement/over generalizing. Creating few recombinant cells and viruses (and/or the down-regulation of gene expression) using CRISPR/Cas9 does not equate to development of new “engineering tools”. I suggest re-writing this sentence to remove this ambiguity.
Figure1: Figure 1A is not clear. The legends are not properly described; therefore, it is difficult to understand the figure. For example, I would place the full column “:” in front of cell line (Cell line:)
Line253-254: is it not expected that GFP will be unaffected in the absence of these molecules?
Lines 343-344: I do not entirely agree with the second part of the sentence “…and protein expression is impacted by translation but not transcription”. This makes the entire sentence and the sentence after it unclear. It is true that an unsuccessful gene disruption or a silent mutation will not affect encoded protein expression and function. But does it also mean that protein expression is not impacted by transcription? ...and for all indel mutations that are silent, or for any unsuccessful targeted gene disruption? Do the authors mean that indel mutations resulting from NHEJ faulty (non-homologous end joining) repairs of Cas9 DNA strand breaks always (?) result in silent mutations in which observed transcriptional changes do not impact protein translation? This is also not correct because not all indel result in silent mutations.
Author Response
Reviewer 1:
Comparison of CRISPR-Cas9 tools for transcriptional repression and gene disruption in the BEVS
 In this work, Bruder et al showed that Cas9 may be more effective than dCas9 for total gene disruptions, whereas dCas9 allows gene expression albeit at reduced level relative to the wildtype. Bruder et al used the Baculovirus as a model organism for the study.
This is an interesting study with good experimental plans. Adequate controls were employed to clearly show the impacts of the CRISPR/Cas9 tools. The manuscript is well written in a simple-to-understanding language without compromise to scientific facts/data. Findings from this study will be useful to researchers using Baculovirus vector systems (BEVs)
Other comments
Line 81: “serum-free” not “serum free”
Response: Thank you for pointing out this error; it has been corrected in the text.
Lines 67-70: This study no doubt makes an original contribution to the field of BEVs. However, I find the statement “Here, we have extended upon these recent advancements by developing engineering tools based on Cas9 and its nuclease deficient variant dCas9 for targeted gene disruption…” quite an over statement/over generalizing. Creating few recombinant cells and viruses (and/or the down-regulation of gene expression) using CRISPR/Cas9 does not equate to development of new “engineering tools”. I suggest re-writing this sentence to remove this ambiguity.
Response: We thank the reviewer for pointing out this concern and we understand their objection. We have revised this sentence to read:
Here, we have extended upon these recent advancements by implementing and comparing strategies based on Cas9 and its nuclease deficient variant dCas9 for targeted gene disruption (CRISPRd) and transcriptional repression (CRISPRi), respectively
Figure1: Figure 1A is not clear. The legends are not properly described; therefore, it is difficult to understand the figure. For example, I would place the full column “:” in front of cell line (Cell line:)
Response: We thank the reviewer for pointing this out and we agree that the legend is not clear. We have made the following revision to the legend for this figure:
(215-222) QPCR Ct values of the Sf9-encoded 28S rRNA housekeeping gene, (d)cas9 gene, and virus-encoded vp39 and p6.9 promoter-expressed gfp reporter gene in baculovirus-infected Sf9 (squares), Sf9-Cas9 (circles), and Sf9-dCas9 (triangles) cells at 0, 24, 48, and 72 hpi. B. GFP fluorescence intensity of baculovirus-infected Sf9 (squares), Sf9-Cas9 (circles), and Sf9-dCas9 (triangles) cells at 24, 48, and 72 hpi and C. infectious virus titer of supernatants from infected Sf9, Sf9-Cas9, and Sf9-dCas9 cells at 72 hpi.
Line253-254: is it not expected that GFP will be unaffected in the absence of these molecules?
Response: Yes, this result is expected. The statement was meant to convey that the result is consistent with CRISPR-mediated targeting (as opposed to an aberrant effect from expression of cas9 or the sgRNA by themselves). We have revised this sentence to make this more clear by adding:
(259) … “and is therefore consistent with CRISPR-mediated targeting of the gfp gene.”
Lines 343-344: I do not entirely agree with the second part of the sentence “…and protein expression is impacted by translation but not transcription”. This makes the entire sentence and the sentence after it unclear. It is true that an unsuccessful gene disruption or a silent mutation will not affect encoded protein expression and function. But does it also mean that protein expression is not impacted by transcription? ...and for all indel mutations that are silent, or for any unsuccessful targeted gene disruption? Do the authors mean that indel mutations resulting from NHEJ faulty (non-homologous end joining) repairs of Cas9 DNA strand breaks always (?) result in silent mutations in which observed transcriptional changes do not impact protein translation? This is also not correct because not all indel result in silent mutations.
Response: We thank the reviewer for bringing our attention to these statements. Our aim was to discuss these results in the context of the different mechanisms of CRISPR-mediated transcriptional repression versus gene disruption; transcriptional repression actually reduces the number of mRNA molecules of the target gene (which would result in less protein abundance assuming the rate of translation or protein per mRNA molecule is constant). For CRISPRd, however, the knockout is due to introduction of indels to the DNA. This should not significantly impact transcription (from the view of number of mRNA transcripts produced) but the translated protein is non-functional (except in the case of silent mutations). We have revised these statements to hopefully help with clarity:
(347-355) This ultimately leads to an overall, but typically incomplete, reduction in fluorescence intensity without introducing irreversible mutations in the genome. On the other hand, gene disruption mediated by CRISPRd results in indel mutations from dsDNA break repair which are irreversible and often result in loss-of-function mutations in the encoded protein[28,29]. Consequently, gene copies with indel mutations in the ORF are transcribed but the encoded protein is non-functional, whereas gene copies that are not successfully targeted or the indel mutation is silent would be transcribed and functional protein is translated at wildtype levels.
Reviewer 2 Report
Here in this interesting paper, Bruder et al., aim to develop a knock-out system based in the utilization of the CRISPR systems for the disruption and transcriptional repression in the baculovirus expression vector system (BEVS). The authors serve in the utilization of two CAS9 variant; the active for the complete gene deletion and the dCAS9 for the transient down regulation. The paper is well written and clear, including appropriate controls, however there is some minores consideration to be considered before publication.
- They develop first a Sf9 line expressing either Cas9 and dCAS9, and they demonstrate that the cell line express Cas9 and dCAS9. I have a concern about the decrease observed from 24 to 72 h post infection, in our hand when a cell line is established, we observe a steady level of gene expression over the time, may the authors explain more in detail, why we observe such decrease, either in the expression of Cas9 and dCAS9 in the established cells lines.
- The author disrupts GFP using the developed systems, is really surprising to see in Figure 2 how the % of eGFP positive cells decrease dramatically 24 HPI, having in account that the eGFP is a very stable protein! This may be probably because the Sf9 cell cycle is verry active and therefore the elimination of eGFP is verry affective ¿may I ask the authors to give more explanation!!? In this point, may a cycle experiment clarify any doubt?? Thank you! On the other as in other figures, the authors do not indicate the type of statistical analysis that has been used (applicable to the whole paper).
- In FIG 3 the authors show the disruption of two viral genes IE-1 and vlf-1, and they measure eGFP expression and viral progeny. Once again, the expression of eGFP drop immediately, 24-hour post infection, and the expression increase either in the case of CAS9 and dCAS9, the late in normal to expect such increase, but not in the case where CAS9 is used. Pleas can the authors explain more in details these data.
- It is very useful and clarify for the reader and reviewers, if a figure with a cartoon showing the different construct used in the study and the genome editing strategies was included in the papers (maybe as SM).
Author Response
Reviewer 2:
Here in this interesting  paper, Bruder et al., aim to develop a knock-out system based in the utilization of the CRISPR systems for the disruption and transcriptional repression in the baculovirus expression vector system (BEVS). The authors serve in the utilization of two CAS9 variant; the active for the complete gene deletion and the dCAS9 for the transient down regulation. The paper is well written and clear, including appropriate controls, however there is some minores consideration to be considered before publication.
 
- They develop first a Sf9 line expressing either Cas9 and dCAS9, and they demonstrate that the cell line express Cas9 and dCAS9. I have a concern about the decrease observed from 24 to 72 h post infection, in our hand when a cell line is established, we observe a steady level of gene expression over the time, may the authors explain more in detail, why we observe such decrease, either in the expression of Cas9 and dCAS9 in the established cells lines.
Response: We thank the reviewer for raising this concern. Downregulation of many host genes is commonly observed in baculovirus-infected insect cells (for example, see PMID 29432775 or 28527118). Recent evidence from a different baculovirus (BmNPV) has suggested a prominent role of epigenetic regulation mediated by the virus (PMID 32511266). This downregulation appears to happen even for cells transformed with expression cassettes using immediate-early viral promoters over the course of the infection cycle (for example see the IE1 promoter SEAP RNA in fig 2 of PMID 23458965 at 24, 48 and 72 hpi). Consequently, we weren’t surprised to see the downregulation of either cas9 gene variants over the course of the infection in our results. Nevertheless, these initial experiments to characterize expression of Cas9/dCas9 in our developed cell lines was important to report in the manuscript.
- The author disrupts GFP using the developed systems, is really surprising to see in Figure 2 how the % of eGFP positive cells decrease dramatically  24 HPI,  having in account that the eGFP is a very stable protein! This may be probably because the Sf9 cell cycle  is verry active and therefore the elimination of eGFP is verry affective ¿may I ask the authors to give more explanation!!? In this point, may a cycle experiment clarify any doubt?? Thank you! On the other as in other figures, the authors do not indicate the type of statistical analysis that has been used (applicable to the whole paper).
Response: We thank the reviewer for these comments. Statements regarding statistical analyses were added to relevant subsections in the materials and methods (lines 152-154, 168-170, and 198-199).
Regarding the GFP fluorescence data presented in Figure 2: the % GFP positive of the gfp-targeted viruses (GFP2, GFP3, and GFP4) is significantly lower compared to the control virus at 24 hpi, particularly for infections in Sf9-Cas9 cells. These results are consistent with CRISPR-mediated prevention of transcription of the gfp gene (for CRISPRi) and production of functional GFP protein (for CRISPRd). Between 24 and 48 hpi, we did see an increase in the % GFP positive population (for example the GFP4 virus, shown in the lightest gray coloured bars). This increase between 24 and 48 hpi is most likely due to small differences in the timing of infection cycle events between cells. A MOI of 3 is used in order to result in simultaneous infection of every cell with at least 1 virus particle, but pragmatically we know that this is a distribution of time as well as number of particles entering a given cell. This also results in slight differences in the timing of beginning of transcription of the virally encoded gfp gene and thus a distribution in fluorescence intensity levels, as well. At 24hpi, there are likely some events that have fluorescence intensities that are similar to the negative control and so get counted as negative events but by 48hpi there has been enough expression of GFP such that the flow cytometry fluorescence intensity measurement is higher than the negative control and is therefore GFP-positive. It is therefore likely that the 48hpi time point is likely the best represents the efficiency of the CRISPR approach (for example, for Sf9-Cas9 cells, our results suggests that the GFP3 target resulted in complete disruption of the gfp gene in approximately 50% of cells rather than the roughly 70% of GFP negative cells that we observed at 24hpi.)
- In FIG 3 the authors show the disruption of two viral genes IE-1 and vlf-1, and they measure eGFP expression and viral progeny. Once again, the expression of eGFP drop immediately, 24-hour post infection, and the expression increase either in the case of CAS9 and dCAS9, the late in normal to expect such increase, but not in the case where CAS9 is used. Pleas can the authors explain more in details these data.
Response: We thank the reviewer for the comment and would refer back to the response from pt 2 above as that explanation applies to this comment as well.
- It is very useful and clarify for the reader and reviewers, if a figure with a cartoon showing the different construct used in the study and the genome editing strategies was included in the papers (maybe as SM).
Response: We thank the reviewer for this suggestion; however, since only the promoter used for transcription of the gfp reporter gene and the specific sgRNA spacer sequence used for (d)Cas9 targeting changed during these experiments, we feel that a figure showing the different constructs is unnecessary. If the reviewer feels strongly about this point, however, we will happily oblige their request.